∂ | **Open Peer Review** | Microbial Genetics | Research Article

# Global genomic diversity and conservation of SARS-CoV-2 since the COVID-19 outbreak

Heng Li,[1,2,3] Liping Ding,[1] Rui Liao,[1] Nini Li,[4] Xiaoping Hong,[1] Zhenyou Jiang,[5] Dongzhou Liu[1]

**ABSTRACT** Global sequencing of the severe acute respiratory syndrome coronavirus 2 (SARS-CoV-2) has enabled researchers to monitor the genetic information of new pathogenic variants. However, the integrity and accuracy of most genomic information cannot be guaranteed due to limited sample quality and sequencing technology. This restricts researchers from tracking the latest changes in the virus genome. In this study, we aimed to characterize the diversity and conservation balance of complete SARS-CoV-2 genomes worldwide since the outbreak. To achieve this goal, we collected 5,966,490 genome sequences from various parts of the world, excluding those with incomplete or unknown/degenerate sequences. Our methodology included comparing sequences using BLASTN and BLASTP, analyzing RNA secondary structure, and calculating entropy. Our findings provide insights into the characteristics of SARS-CoV-2 and potential mechanisms of pathogenesis. We found that uracil had the highest proportion of all bases among various coronaviruses and cytosine to uracil mutations had the highest proportion among all point mutations. The consistency in the front part (1–26,599 nt) was significantly higher than that in the back part (26,600–29,903 nt) of the genome. For most genes, the similar consistency characteristics were also observed in their protein families.

**IMPORTANCE** Our results indicate that most severe acute respiratory syndrome coronavirus 2 genomes sampled from patients had a mutation rate ≤1.07 ‰ and genome-tail proteins (including S protein) were the main sources of genetic polymorphism. The analysis of the virus-host interaction network of genome-tail proteins showed that they shared some antiviral signaling pathways, especially the intracellular protein transport pathway.

**KEYWORDS** coronavirus, SARS-CoV-2, genomic diversity, COVID-19

A s of January 2023, the coronavirus disease 2019 (COVID-19) pandemic caused by the severe acute respiratory syndrome coronavirus-2 (SARS-CoV-2) has resulted in a staggering 655 million confirmed cases and 6.67 million deaths worldwide (https://covid19.who.int/). Despite the rapid progress in related research on SARS-CoV-2, driven by the combined efforts of healthcare facilities and researchers, the extensive infections have led to the emergence of many variants (1, 2). Currently, there are 29 clades or 35 emerging lineages, including variants of concern (VOC), variants of interest (VOI), and variants under monitoring (VUM) (https://gisaid.org/database-features/influenza-genomic-epidemiology/). In recent years, there have been several rounds of mainstream strain replacement, with one after another emergence of original strains that have replaced all parts of the early years, including Alpha GRY (B.1.1.7) (3), Delta GK (B.1.617.2 + AY.x) (4, 5), and Omicron GRA (B.1.1.529 + BA.x) (6, 7). Although some strains have been dominant over months, there is still an urgent need to track and study all mutants to help

Address correspondence to Dongzhou Liu, liu_dz2001@sina.com, or Heng Li, lconstant@foxmail.com.

The authors declare no conflict of interest.

See the funding table on p. 13.

predict unknown strains and reveal the evolutionary laws of SARS-CoV-2 to cope with its rapid genetic evolution.

RNA viruses, like SARS-CoV-2, undergo frequent genetic changes that pose significant challenges to human health in terms of disease diagnosis, prevention, and treatment. Nucleic acid test is currently the most accurate diagnostic method, but mutations in viral genes may lead to mispriming or mismatching of the detection primers, resulting in false negative results (8–10); rapid antigen tests are also less sensitive to some variants (11). The mutation of the spike protein helps variants, including Omicron, adapt to and escape from the human immune system, affecting the effectiveness of immunity and treatment (12, 13). As vaccines are effective means of preventing diseases, viral gene mutations may lead to immune escape (14). Drug resistance may occur when monoclonal antibodies against SARS-CoV-2 encounter variants, for example, monoclonal antibodies Ly-CoV555 and REGN10933 have impaired or lost activity against B.1.526-E484K strain (15); similarly, the neutralizing activity of monoclonal antibodies against B.1.351, B.1.1.28, B.1.617.1, and B.1.526 viruses was reduced or eliminated in cell culture (16). Many studies have investigated key variants or proteins, including S protein (the major antigen), or a smaller number of genomes (17–19). However, due to the limitation of sample quality and sequencing technology, there is a lack of large-scale complete genome research. Continuous whole-genome sequencing of SARS-CoV-2 is necessary to maintain genomic surveillance (20); further analysis and characterization of the novel variants are critical to understanding the prevention and therapy of COVID-19.

In this study, we attempted to identify conserved and essential RNA secondary structures in the genome of SARS-CoV-2, with a particular emphasis on completeness and accuracy, which are critical for developing targeted therapies to combat viral mutations. To achieve this goal, we compiled SARS-CoV-2 sequence data from 5,966,490 samples globally, focusing only on complete genomes and excluding any uncertain base sequences. Then, the original proportions of the different bases and the base changes brought about by the variants were investigated. Following the alignment of all complete genomes with the full length of 29,903, we analyzed the identity of bases at each position and calculated the identity rate per 100 nt. Moreover, we investigated genes with significant differences and their variations in comparison to homologous proteins across different viruses. We also identified human key interacting proteins and their common pathways that correspond to SARS-CoV-2.

## RESULTS

### Global SARS-CoV-2 genome composition and distribution

As of 22 January 2023, the SARS-CoV-2 Data Hub contains 5,966,490 nucleotides in the NCBI Virus database. After excluding incomplete sequence information, 1,394,180 complete genome sequences were obtained. As the length of the reference genome NC_045512.2 is 29,903 nucleotides, 80,833 genomes were selected as candidates based on the length. These candidate samples come from various parts of the world, with the highest number in Europe, followed by North America and Asia (Fig. 1). These data represent sequencing data only, not the total number of COVID-19 cases.

### Uracil has the highest proportion in the SARS-CoV-2 genome, and cytosine to uracil mutations are the main type of point mutations

Examination of the 80,833 genome sequences revealed 22,588 duplicate values, leaving 58,245 unique values after removal. Subsequently, genomes with any missing (–) and degenerate information (R, Y, K, M, S, W, B, D, H, V, and N) were removed, resulting in 5,966 genomes with well-defined sequences. When analyzing these genomes as a whole, the highest proportion was U at 32%, followed by A > G > C (Fig. 2A). Surprisingly, despite C being the lowest (18%), the proportion of cytosine to uracil (C-to-U) mutations was the highest of the 12 types of point mutations (36%). The second was G-to-U (16%). The other ten mutations were between 2% and 9%. A staggering 58% of mutations were converted to U, while only 11% were converted to C (Fig. 2B and C).

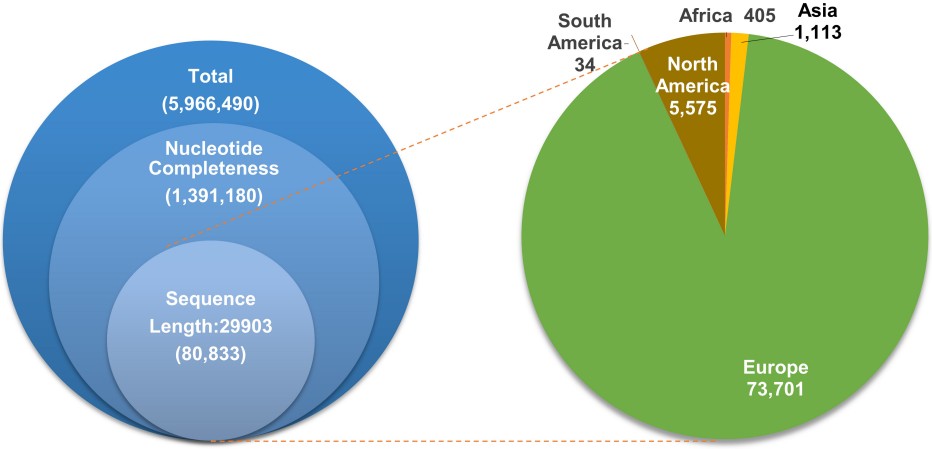

**FIG 1** Screening and distribution of the SARS-CoV-2 genome.

We compared the genomes of seven human-infecting coronaviruses, including SARS-CoV-2, and found that their sizes were similar (27,277 to 30,713 nt), with SARS-CoV-2 ranking third in size after human coronavirus OC43 (HCoV-OC43) and Middle East respiratory syndrome coronavirus (MERS-CoV) (Fig. 3A). Base composition analysis showed a consistent order, U > A > G > C, in all seven viruses (Fig. 3B).

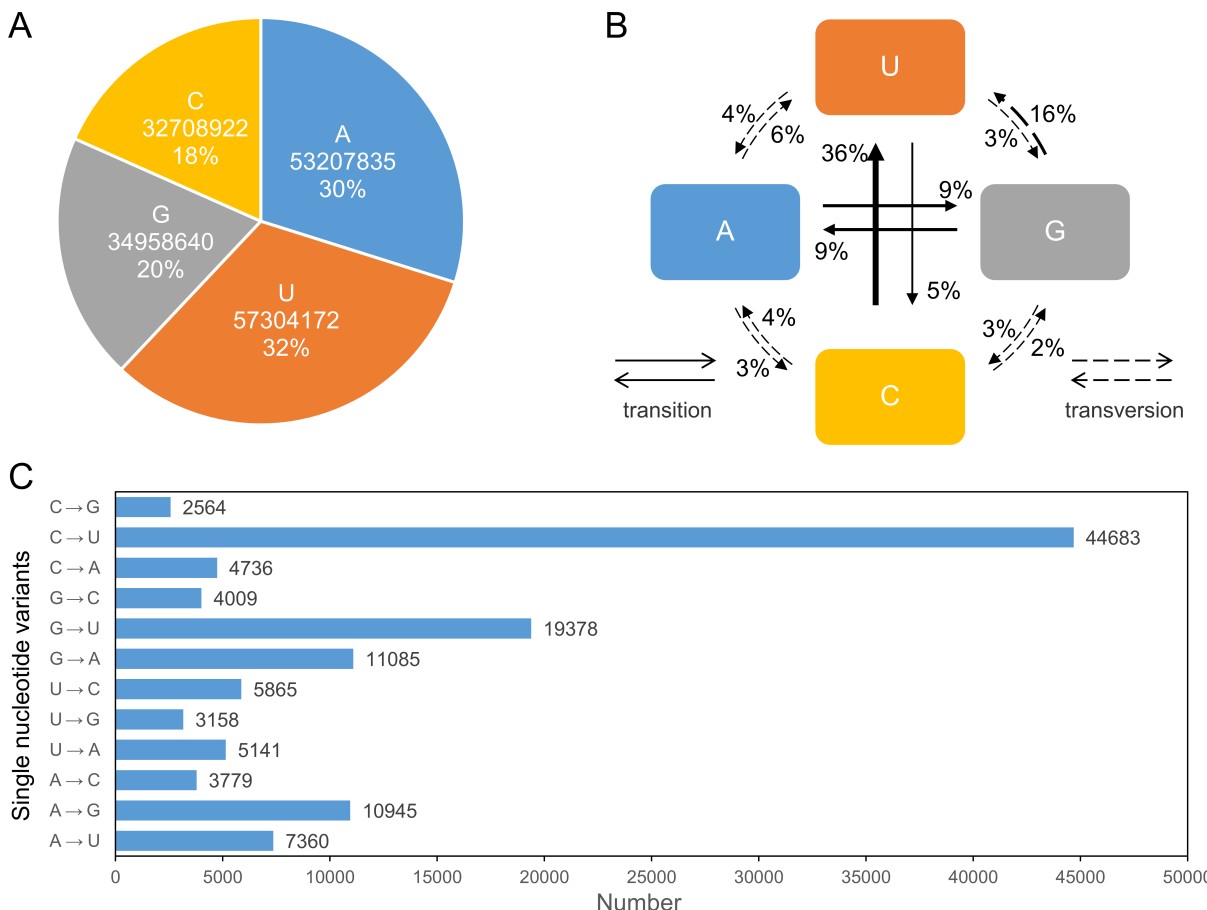

**FIG 2** (A) The proportion of the four bases in the SARS-CoV-2 genome. (B) The proportion of various transitions and transversions in point mutations. (C) The number of 12 types of point mutations.

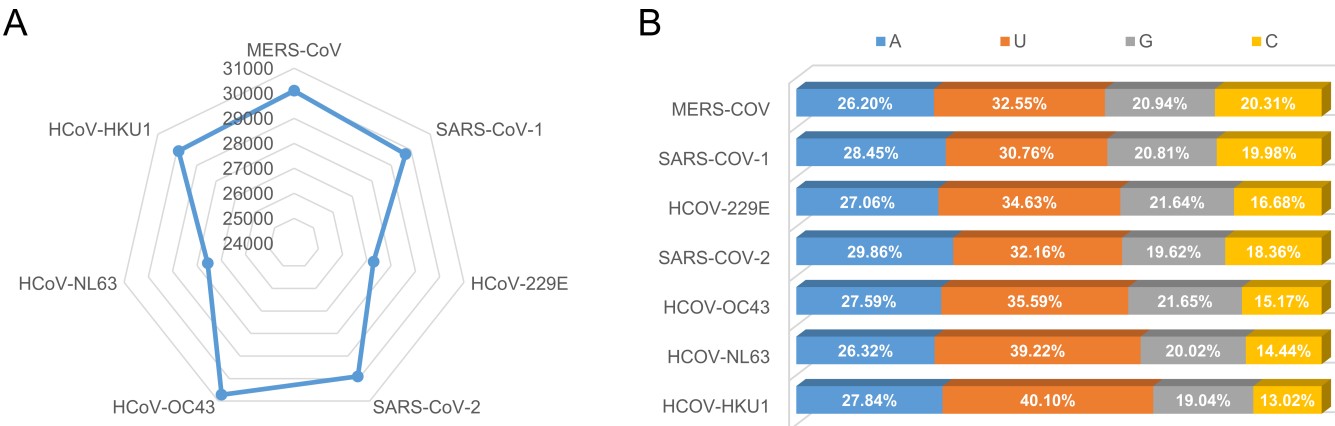

FIG 3   (A) The reference genome length of coronaviruses that can infect humans. (B) The ratio of bases in the different coronaviruses.

## The mutation rate of most variants is less than 1‰ (median 0.60‰)

The genomic sequences of the variants exhibit some diversity. We aligned the 80,833 genomic sequences, and there were 295 genomes with more than 40 mismatches to the consensus sequence, of which the top three have 15,494, 15,492, and 2,623 mismatches, respectively (Fig. 4A). Considering that 15,494 and 15,492 are close, we performed sequence alignments and found that they only have two different nucleotides (Fig. 4B). After alignment to the reference genome NC_045512.2, we found that these two variants had two deletions and one insertion mutation, and the number of deleted and inserted nucleotides were equal, resulting in no change in genome length (3 + 9 = 12; Fig. 4C). Further sequence analysis revealed that the two deletion mutations leading to SS-to-C mutation in Nsp3 and an SGF deletion in Nsp6, respectively. The insertion mutation was located in the non-coding region between ORF8 and the N protein (Fig. 4D). We aligned the insertion sequence "AACAAACAAACA" to the reference genome sequence NC_ 045512.2; the results showed the highest similarity to sequences near the location of insertion mutations (28,259–28,269; Fig. 4E). The above mutations resulted in an unusually high number of mismatches, and after excluding these two variants, the number of mismatches for all the other variants ranged from 0 to 2,623, with a median of 18 (0.60 ‰), and 90% of variants had less than 30 (1.07 ‰).

The insertion mutation causing "AACA" change to "AACA-AACA-AACA-AACA" occurred in the non-coding region between ORF8 and N protein (Fig. 5A), seven nucleotides away from the start codon of the N protein. While this mutation does not affect the protein sequence, its impact on the Shine-Dalgarno (SD) sequence is unknown. We simulated the RNA secondary structure before and after the mutation (Fig. 5B) and found that the insertion mutation led to a larger loop in front of the start codon, which may reduce complementarity with the ribosome and slow down its binding speed, thereby affecting translation rate. However, there were no obvious structural changes upstream and downstream of the mutation. Deviations from the baseline in the entropy plot indicate that the structure involving the mutant has low positional entropy, whereas the original sequence of the downstream gene affected by the mutation has higher randomness and positional entropy (dashed lines label the mutation).

## The consistency of the front part was significantly higher than that of the tail part in the SARS-CoV-2 genome

After all genome sequences were aligned, 1–29,903 bases were analyzed sequentially. If the sequence of all genomes in the same location is identical, it is defined as identity; otherwise, it is defined as mismatch. The percent identity was 55.98% for the whole genome (Fig. 6A). Subsequently, continuous identical sequences were analyzed. As the length increased, the number of continuous sequences decreased (Fig. 6B). There were

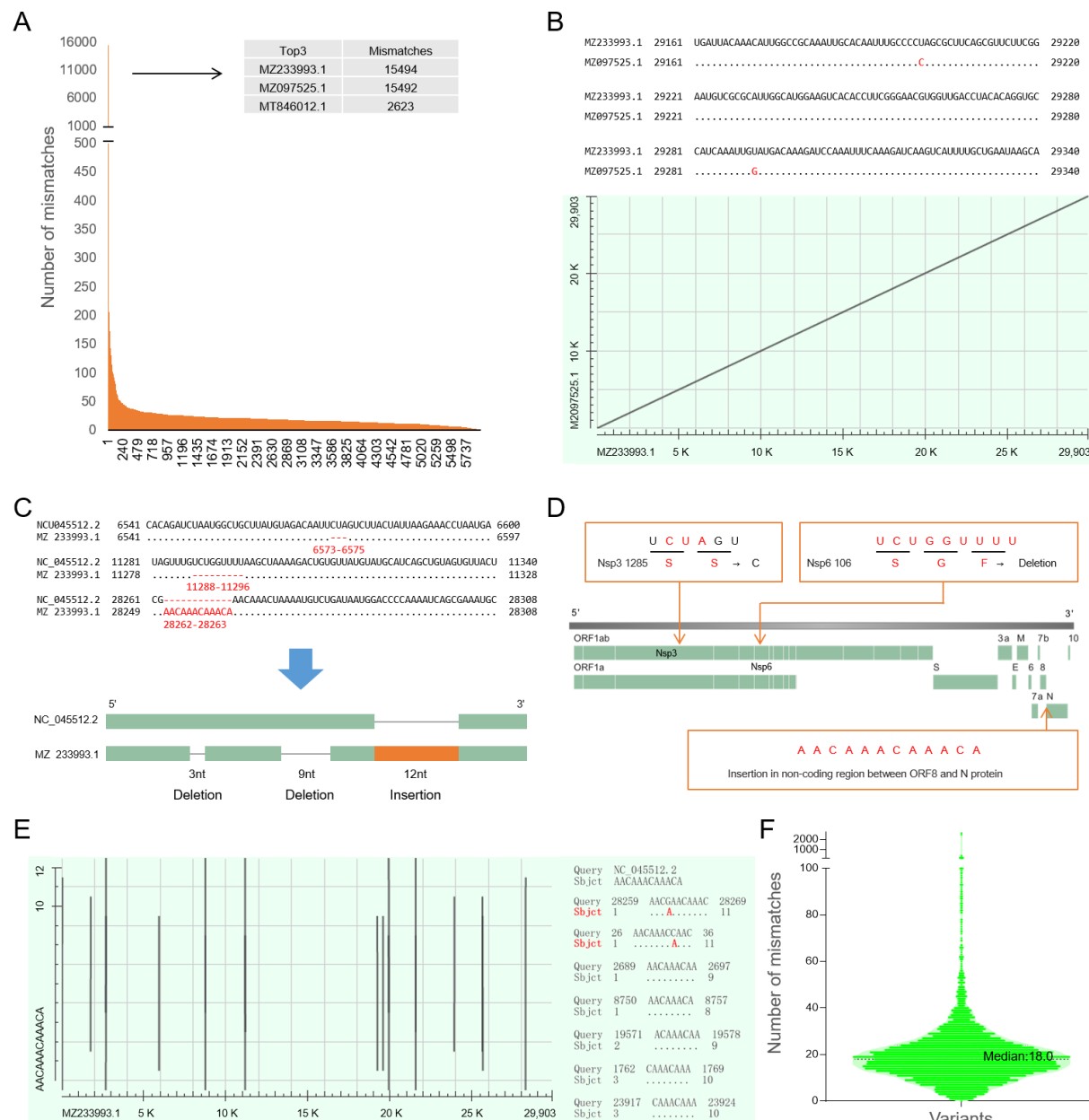

**FIG 4** (A) Number of mismatches in the variant genomes (descending order). (B) The two variants with the highest mismatches (MZ233993.1 and MZ097525.1) were highly similar in sequence, only two bases are different. (C) MZ233993.1 and MZ097525.1 contain two deletion mutations and one insertion mutation, and the number of deleted bases was equal to the number of inserted bases. (D) The deletion mutation in MZ233993.1 and MZ097525.1 results in two or three amino acid mutations, and the insertion mutation is located in a non-coding region between ORF8 and the N protein. (E) The alignment result of the inserted sequence "AACAAAAAACA" and reference genome NC_045512.2 aims to explore the source of insertion mutation. (F) After excluding the frameshift mutation, the overall distribution of the number of mismatches in the variants.

1,225 sequences at $n = 5$. There were only 13 sequences at $n > 20$. The longest continuous consistent sequence length is 32. At the genome-wide level, the percent identities were calculated per 100 nt, from which the genome-wide landscape was mapped (Fig. 6C). The results showed that the consistency of the front part of the genome (1–26,599 nt) was 62.64%, significantly higher than the 4.03% in the tail part (26,600–29,903 nt). The four highest sites are inside the genes Nsp3, Nsp8, RdRp, and S (annotated with blue arrows). The genes located at the end of the genome, particularly those beyond the M gene, exhibit low consistency. Certain genes display significant internal consistency changes, particularly two positions within the S and Nsp14 genes that are as low as 0.

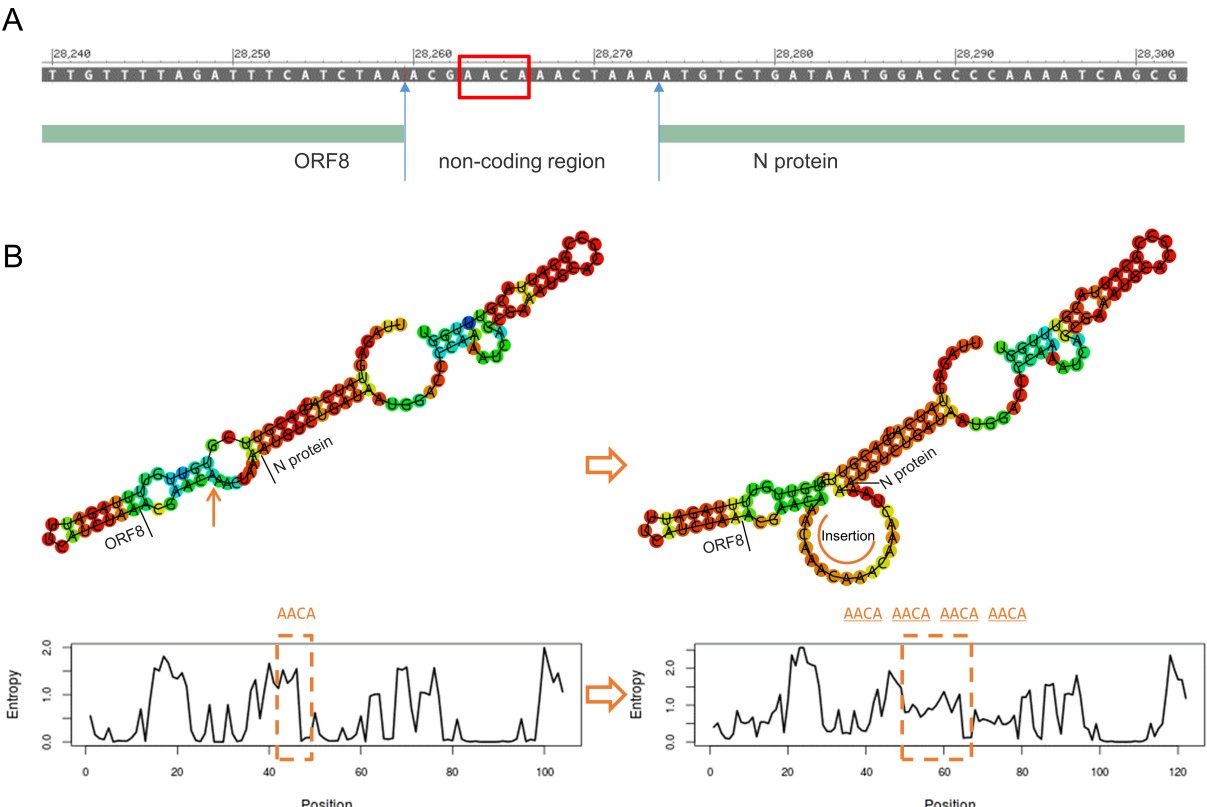

**FIG 5** (A) Position of the insertion mutation. (B) Comparison of the secondary structure and energy entropy between the reference genome NC_045512.2 and the insertion mutation (MZ233993.1 and MZ097525.1).

The consistency of most genes in ORF1ab was higher than 62%, with a median of 69%; the S gene showed large internal differences; the majority of genes in the tail were less than 13%, with a median of 6% (Fig. 7A). The top four sites with the highest consistency in SARS-CoV-2 also showed high conservation in their respective protein families (Fig. 7B). Among them, the sites in proteins RdRp and S are also highly conserved in their protein families. There are one and three different amino acid sequences in Nsp3 and

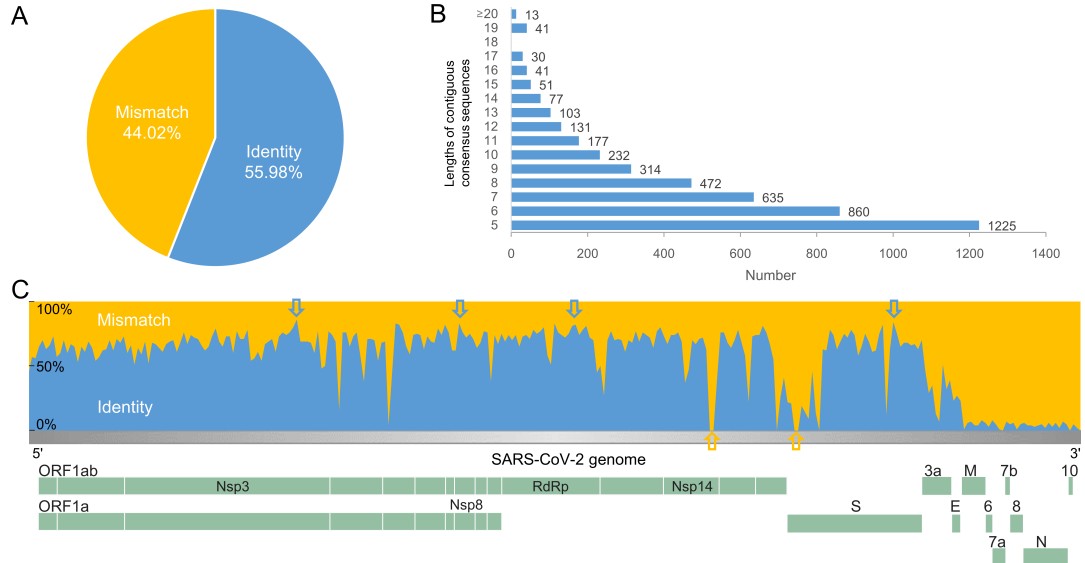

**FIG 6** (A) The percent identity in the SARS-CoV-2 genome. (B) Continuous identical sequences in the SARS-CoV-2 genome.

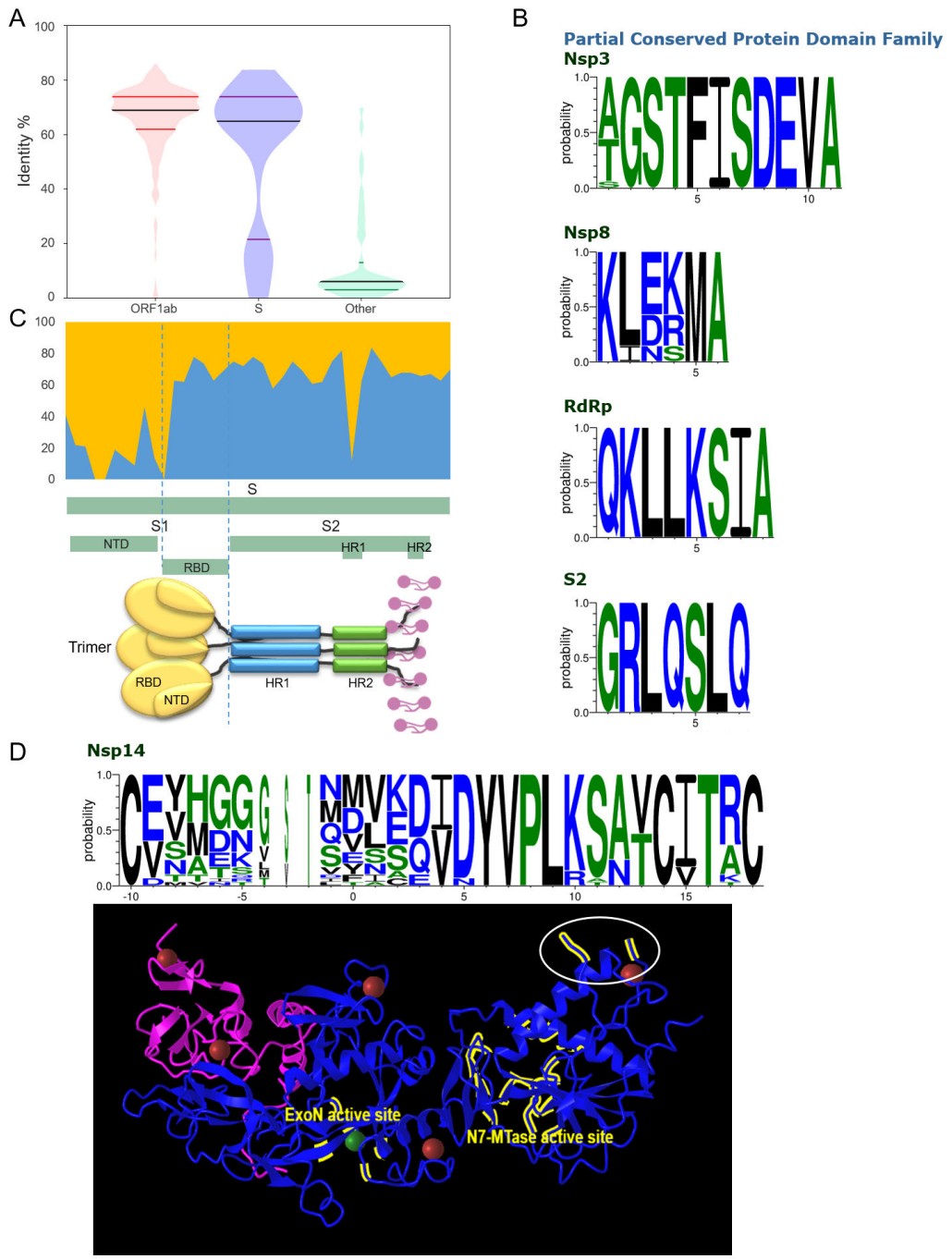

**FIG 7** Analysis of several gene loci with the largest differences in consistency. (A) Consistency analysis of the anterior part of the genome (ORF1ab), S gene, and the tail of the genome (other genes). (B) Conservation analysis of complete consensus sequence across protein domain families. (C) The relationship between sequence consistency and different domains in S protein. (D) Conservation and structural analysis of the least conserved site in SARS-CoV-2.

Nsp8, which can be used as important sites for virus identification. S protein contains multiple domains; S1 plays a role in the binding of receptor ACE2 (21), whereas S2 mediates the membrane fusion process (22). Partial domains and sequence identity are

highly correlated; the NTD domain in S1 and HR1 in S2 showed a significantly reduced (16.9% and 52.3%), while the consistency of other parts was 69.2%; they showed clear demarcation (Fig. 7C). In addition to S protein, another protein that shows drastic internal variation is Nsp14, most of which had high consistency, but one site was 0. This site not only varies greatly among different SARS-CoV-2 variants but is also extremely poorly conserved among homologous proteins of different viruses. The structure of Nsp14 protein shows two active sites corresponding to two functions; the site is located at a marginal position (shown by white ovals; Fig. 7D).

To further investigate genes with low conservation at the 3′ end of the SARS-CoV-2 genome, we selected five key proteins ORF6, ORF7a, ORF7b, ORF8, and ORF10. A total of 213 corresponding human interacting proteins were collected through literature mining (Table 1). Analysis of these proteins revealed partial overlap or shared pathways, with ORF8 having the highest number of interacting proteins (Fig. 8A). Pathway and process enrichment analysis was performed using DAVID (23) and KOBAS (24) with the following ontology sources: KEGG Pathway, GO Biological Processes, and Reactome Gene Sets ($P < 0.01$, count > 3, and enrichment factor >1.5). We obtained five sets of terms, with the number ranging from 3 to 17 (Fig. 8B). The most significant term for each group was GO: 0035966: response to topologically incorrect protein (ORF8); GO: 0006886: intracellular protein transport (ORF7A); R-HSA-9679506: SARS-CoV infections (ORF6); WP4860: Hijack of ubiquitination by SARS-CoV-2 (ORF 10); and GO: 0006886: intracellular protein transport (ORF 7b). GO: 0006886: intracellular protein transport was a common term among the five groups. To further understand the relationships between the terms, we generated a network diagram (Fig. 8C and D) with different colors representing different groups and terms. The results showed that ORF8 had the widest distribution and was linked to other groups. The size of the node was proportional to the total number of

**TABLE 1** Key proteins with low consistency in SARS-CoV-2 genome tail and their interacting proteins reported in the literature

| SARS-CoV-2 protein | Human interacting proteins | Reference |
|---|---|---|
| ORF6 | NUP98, RAE1, MTCH1, KINH, EDC4, DCTN2, ATD3A, RAE1L, FAF2, XP01, SPF27, MYCB2, AT1A1, ATD3B, AT2A2, ACLY, DESP, ATPB, RT27, ATPA, CE170 | 25 |
| ORF7a | HEATR3 | 25 |
| | MDN1, HACD3, FAF2, XPO1, COPG2, COPB2, SMC2, MIC60, BAG6, QPCTL, XPOT, PSMD1, AT2A2, PP6R3, COPA, PSMD2, PRS7, TECR, COPB, PSMD7, AT1A1, PRS4, DNJA2, UBR5, PSD13, PSMD6, HSP74, SC16A, CMC1, MPCP, GEMI4, PSA, DNJC7, RPB2, PRS8, GHC1, CALU, EMD, DNJA3, TXTP, BAG2, DDX20, DJC10, ECHA, ATPB, 2AAA | 26 |
| ORF7b | HACD3, RO60, XPO1, COPG2, COPB2, M2OM, FAF2, QPCTL, COPA, AT2A2, MIC60, TECR, SMC2, XPOT, TMM33, COPB, MPCP, PSA, AT1A1, ATD3A, SND1, NOC3L, ATPB, DNJA1, ATPA, CALU, DPM1, RCN2 | 26 |
| ORF8 | PLOD2 | 26 |
| | TOR1A, STC2, PLAT, ITGB1, CISD3, COL6A1, PVR, DNMT1, LOX, PCSK6, INHBE, NPC2, MFGE8, OS9, NPTX1, POGLUT2, POGLUT3, ERO1B, PLD3, FOXRED2, CHPF, PUSL1, EMC1, GGH, ERLEC1, IL17RA, NGLY1, HS6ST2, SDF2, NEU1, GDF15, TM2D3, ERP44, EDEM3, SIL1, POFUT1, SMOC1, PLEKHF2, FBXL12, UGGT2, CHPF2, ADAMTS1, HYOU1, FKBP7, ADAM9, FKBP10, HACD3, BAG6, COPG2, XP01, SMC2, MIC60, COPA, AT2A2, CMC1, PSMD1, COPB, TECR, M2OM, XPOT, PRS7, UBR5, AT1A1, PSMD2, DNJA2, ATD3A, SND1, RPB2, SC16A, MPCP, PSD13, PSMD6, DNJA1, GHC1, TF3C1, CALU, GEMI4, PSA, PSA7, GNAS1, ECHA, ATPA, PRS8, ATPB, DDX20, EMD, EIF3A | 25 |
| ORF10 | PPT1 | 26 |
| | CUL2, MAP7D1, THTPA, ZYG11B, TIMM8B, RBX1, ELOC, ELOB, SMC2, MIC60, XP01, FAF2, HACD3, AT2A2, COPG2, AT1A1, IMDH2, XPOT, SC16A, COPA, COPB, GEMI4, YTHD2, MPCP, RCN2, TCPE, RSSA, TCPD | 25 |

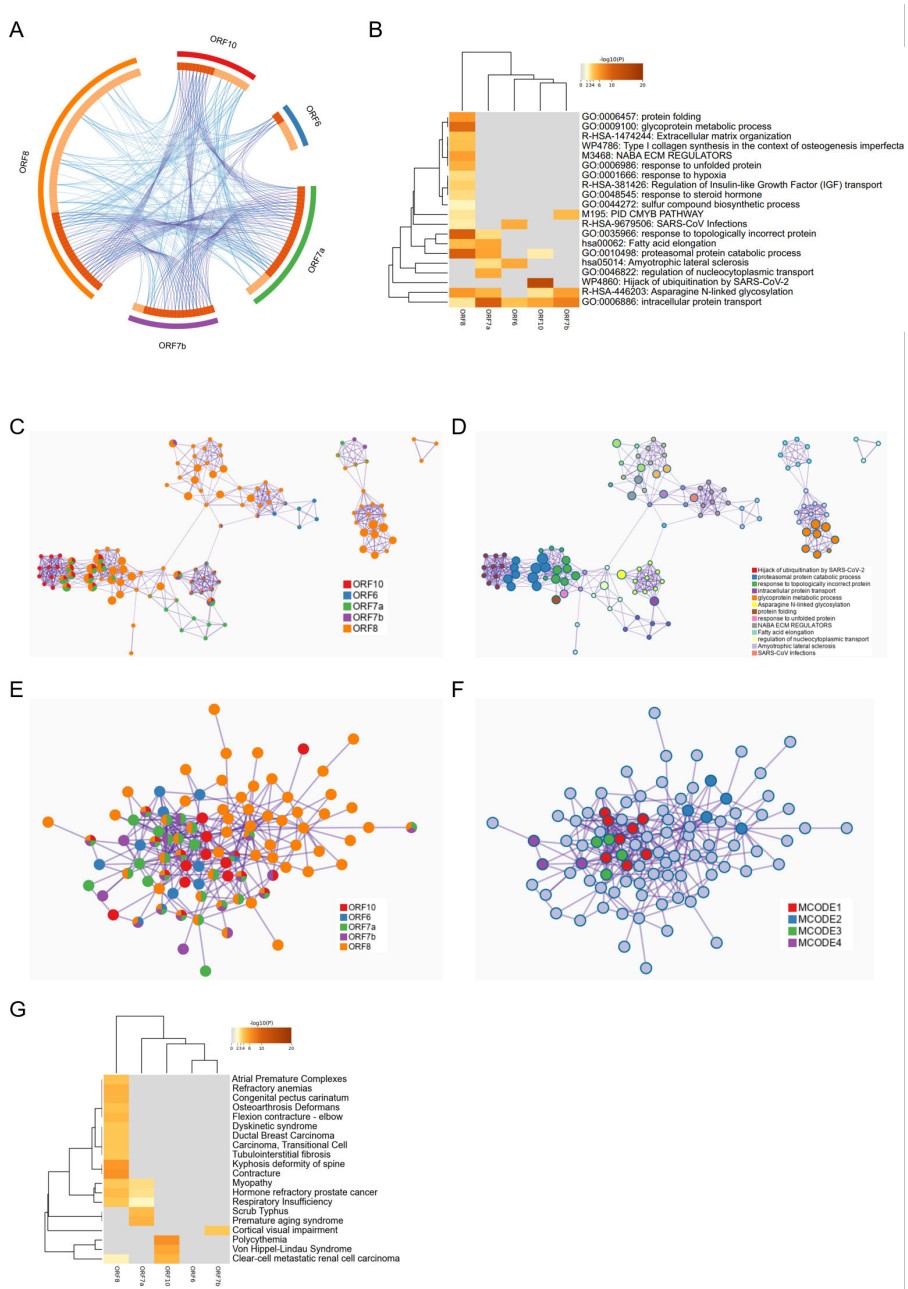

**FIG 8** Genome-wide annotation and analysis of key proteins with low consistency at the C-terminus. (A) The repetition and connectivity among the human proteins interacting with ORF6, ORF7a, ORF7b, ORF8, and ORF10. The blue lines indicate the same protein, while the purple lines represent proteins in the same pathway. (B) Pathway and process enrichment analysis of human proteins interacting with the five key proteins. (C) Enriched term network diagram colored by input clusters. (D) Enriched term network diagram. (E) Enrichment analysis of protein-protein interactions. (F) MCODE components obtained from enrichment analysis of protein-protein interactions. (G) Summary of enrichment analysis performed in DisGeNET.

hits belonging to that particular term, with the proteasomal protein catabolic process, response to topologically incorrect protein, intracellular protein transport, and glycoprotein metabolic process having higher hit rates.

We used STRING (27) (physical score > 0.132) to generate a protein-protein interaction enrichment network, which included subsets of proteins that physically interacted with at least one other member in the list (Fig. 8E). The Molecular Complex Detection

(MCODE) algorithm (28) was used to identify densely connected network components if the network contained 3 to 500 proteins (Fig. 8F). ORF8 had a relatively large proportion and had significant overlap with ORF7a. Hypoxia (MCODE1) and endoplasmic reticulum stress (MCODE2) occupied central positions. Finally, we used the DisGeNET database (29) to help determine the priority of genotype-phenotype relationships (Fig. 8G). Contracture, premature aging syndrome, cortical visual impairment, and polycythemia were identified as symptoms or diseases related to ORF8, ORF7a, ORF10, and ORF7b.

## DISCUSSION

In this work, we reported the diversity and conservation of SARS-CoV-2 genomes worldwide since the outbreak of the COVID-19 pandemic. A wide time range, samples from around the world, complete genome, and accurate sequence are the criteria for selecting samples in this study. Researchers have discovered multiple SARS-CoV-2 epidemic variants, some of which are more successful in evading the immune system's defense, posing challenges for healthcare professionals to control the wave of infection, including VOCs that are gradually replacing original strains around the world. For example, the D614G mutation contributes to the adaptive advantage of SARS-CoV-2 and enhances binding to the human cell-surface receptor angiotensin-converting enzyme 2 (ACE2) (30–32) and is therefore associated with enhanced infectivity. Whereas mutations L452R, S477N, and E484K may result in decreased antibody-neutralizing activity or increased resistance to vaccine-elicited antibodies (33, 34). However, most reports and subsequent studies have focused on mutations in single amino acids or proteins; the horizon cannot be at the genome-wide level.

Completeness and accuracy are two major aspects that limit genetic information. Firstly, due to limitations in sample quality and sequencing technology, many genome sequences are incomplete, with missing regions often occurring at the beginning and end of the sequence. While the lack of poly A at the tail has minimal impact, the starting sequence may contain important genetic information. Degenerate information in sequencing results is a major factor affecting sequence accuracy, and many genomes that contain unknown sequences are excluded, even if these are short. We selected complete genomes with well-defined sequences and removed duplications and incomplete or degenerate information, resulting in 5,966 unique samples for further analysis; these genomes were sourced from around the world, with the majority coming from Europe and the Americas.

We have identified a nucleotide alteration that likely represents an evolutionary hallmark of SARS-CoV-2: C-to-U, which exhibits the highest proportion among all point mutations. Our analysis of the genomes of various coronaviruses that infect humans reveals that uracil is the most abundant among the four bases; this phenomenon is observed in all seven viruses. The primary reason for this alteration may be mediated by apolipoprotein B mRNA-editing enzyme catalytic polypeptide-like (APOBEC) (35). APOBEC was demonstrated in human immunodeficiency virus type 1 (HIV-1), where guanine to adenine (G-to-A) and cytosine to uracil (C-to-U) changes in HIV-1 transcripts have been confirmed (36). However, the related mechanism in SARS-CoV-2 requires further investigation.

We also analyzed the genomic diversity and mutation rate of variants and identified some rare variants with high mismatches, deletions, and insertion mutations. These two variants had no change in genome length despite containing multiple mutations. Although these mutations do not affect protein sequence directly, the insertion mutation may significantly influence the RNA secondary structure by enlarging the loop ahead of the start codon of the N protein, which may affect translational initiation.

The rapid rate of variation is a crucial determinant of RNA viruses to adapt to their living environment; however, a certain level of conservation is a prerequisite for survival. We observed that the anterior segment of the SARS-CoV-2 genome exhibits 62.64% identity, which is significantly higher than the 4.03% identity in the tail segment, suggesting that conservation of the anterior segment is essential for viral

survival, whereas the posterior segment is critical for viral mutation and evolution. Overall, the consistency within most genes is stable, with only a few genes exhibiting large fluctuations. Notably, the identity of the S1-NTD domain in S proteins (16.9%) is significantly lower than that of the adjacent S1-RBD (70.8%). RBD and NTD constitute the two major targets for antibodies (37). The S1-NTD is responsible for binding to human ACE2, and several monoclonal antibodies have been discovered (38). SARS-CoV-2 exhibits improved binding to the ACE2 receptor compared with SARS-CoV (39, 40), implying that S1-RBD and FCS may contribute to the severity of infection in SARS-CoV-2.

The majority of SARS-CoV-2 samples had a genomic mutation rate ≤ 1.07 ‰, with the genome tail and S protein being the main sources of genetic polymorphism. Moreover, we investigated five key proteins ORF6, ORF7a, ORF7b, ORF8, and ORF10, representing the low-conserved genes in the SARS-CoV-2 genome concerning human interacting proteins, pathways, and processes. It showed that they share some antiviral signaling pathways, especially intracellular protein transport.

Although our study provides comprehensive analyses of the complete genome sequence of SARS-CoV-2, several limitations need to be addressed. First, our sample does not include incomplete and degenerate information containing genomes. Therefore, our analysis may not accurately reflect the proportion of mutations in all infected cases. Second, while our investigation emphasizes the importance of non-coding regions in gene regulation, the functional significance of these variants requires further experimental validation. Finally, although we investigated molecular interactions and related diseases using literature mining and databases, more detailed and systematic experiments are needed to prove drug efficacy and side effects.

In conclusion, our study provides valuable insights into the nucleotide composition, mutation profile, genomic diversity, and consistency of the SARS-CoV-2 genome. Our findings enhance our understanding of the virus's features and improve our ability to develop antiviral strategies against the variants. The insertion mutation near the start codon of the N protein may affect translational initiation significantly, further validating the importance of the non-coding regions in gene regulation. Finally, the molecular interactions and related diseases revealed by our investigation of low-conserved genes highlight their biological significance as potential therapeutic targets.

## MATERIALS AND METHODS

### Data source and processing

This study analyzed 5,966,490 nucleotides registered in the SARS-CoV-2 data hub of the NCBI virus database before 22 January 2023 (https://www.ncbi.nlm.nih.gov/labs/virus/vssi/#/). After excluding sequences containing any missing (–) and degenerate information (R, Y, K, M, S, W, B, D, H, V, and N), we obtained 80,833 viral accession (Table S1). During data processing, 22,588 duplicate values and samples with missing or unclear information were removed, resulting in 5,966 clear genome sequences.

### Genomic and protein sequence alignment

Sequence alignment between SARS-CoV-2 genomes was performed using BLASTN, Expect threshold (0.05), word size (7), match/mismatch fraction (1, –1), and gap cost (1, 2) were used as parameters. Low-complexity regions were filtered. BLASTP was used for protein homology analysis. The expected threshold is 0.05, word size is 6, and the cap costs are existence 11 and extension 1.

### Genome consistency analysis

The genome sequence data to be analyzed were downloaded through the NCBI virus database, and the collected sequences were aligned with 1–29,903 nt of the reference genome using alignment software (BLAST or EmEditor) to find out the consistent and

mismatch parts. Dividing the concordant fraction in each 100 nt by the total yielded the consistency rate, and characteristics of the consistency rate were determined.

## Protein structure simulation analysis

The protein sequence data to be analyzed were collected through the public database PDB, and the three-dimensional structure prediction of the protein was performed using protein structure prediction software (SWISS-MODEL) to analyze the simulation results and determine the conformational characteristics, stability, and function of the protein.

## RNA secondary structure analysis and entropy calculation

The 63-nt RNA (TTGTTTTAGATTTCATCTAAACG<u>AACA</u>AACTAAAATGTCTGATAATG6ACCCCA AAATCAGCG), which includes a partial coding sequence for ORF8 and N protein along with a non-coding region in the middle, and its corresponding 75-nt mutant (TTG TTTTAGATTTCATCTAAACG<u>AACAAACAAACAAACA</u>AACTAAAATGTCTGATAATG6ACCCCAAA ATCAGCG), were submitted to the RNAfold Web server (http://rna.tbi.univie.ac.at//cgi-bin/RNAWebSuite/ RNAfold.cgi) using the "minimum free energy (MFE) and partition function" algorithm to avoid isolated base pairs. Other parameters were left as default to generate positional entropy, center of mass structure, and potential MFE structure.

## Gene ontology pathway enrichment analysis

We used the GO annotation tool DAVID for gene functional annotation, which consists of three main aspects: molecular function, cellular component, and biological process. Online tools DAVID was utilized for KEGG and reactome pathway analysis. The reliability and biological significance of the analysis results were evaluated. Statistical tests such as Fisher's exact test were performed to validate the analysis results. The analysis results were presented in the form of graphs, heat maps, and other visualizations using R and Cytoscape.

## Statistical analysis

Statistical analysis was performed using SPSS 17.0 software (SPSS, Chicago, USA). The $t$-test was utilized for group comparisons, and data were presented as mean ± standard deviation. GraphPad Prism 8 was employed to generate graphics. $P < 0.05$ was considered statistically significant.

## ACKNOWLEDGMENTS

This work was sponsored by the Sanming Project of Medicine in Shenzhen (SZSM201512019) and the Research and Development Projects in Key Fields of Guangdong Science and Technology Department (2019B020229001).

This work was funded by the National Natural Science Foundation of China (No. 81971464), the China National Postdoctoral Program for Innovative Talents (BX20200151), and Shenzhen Fund for Guangdong Provincial Highlevel Clinical Key Specialties (No. SZXK011).

## AUTHOR AFFILIATIONS

[1]Department of Rheumatology and Immunology, Shenzhen People's Hospital (The Second Clinical Medical College, Jinan University, The First Affiliated Hospital, Southern University of Science and Technology), Shenzhen, Guangdong, China
[2]Integrated Chinese and Western Medicine Postdoctoral Research Station, Jinan University, Guangzhou, China
[3]Department of Geriatrics, Geriatric Center, Shenzhen People's Hospital (The Second Clinical Medical College, Jinan University, The First Affiliated Hospital, Southern University of Science and Technology), Shenzhen, Guangdong, China

[4]Department of Pathology, Shenzhen People's Hospital (The Second Clinical Medical College, Jinan University, The First Affiliated Hospital, Southern University of Science and Technology), Shenzhen, Guangdong, China

[5]Department of Microbiology and Immunology, School of Medicine, Jinan University, Guangzhou, China

## AUTHOR ORCIDs

Heng Li  http://orcid.org/0000-0001-6750-3911
Dongzhou Liu  http://orcid.org/0000-0003-3766-6562

## FUNDING

| Funder | Grant(s) | Author(s) |
| --- | --- | --- |
| MOST | National Natural Science Foundation of China (NSFC) | 81971464 | Dongzhou Liu |
| China Postdoctoral Foundation Project | National Postdoctoral Program for Innovative Talents (Postdoctoral Innovation Talent Support Program of China) | BX20200151 | Heng Li |
| 深圳市科技创新委员会 | Sanming Project of Medicine in Shenzen Municipality (Sanming Project of Medicine in Shenzhen) | SZSM201512019 | Dongzhou Liu |

## AUTHOR CONTRIBUTIONS

Heng Li, Conceptualization, Funding acquisition, Methodology, Project administration, Supervision, Visualization, Writing – original draft, Writing – review and editing | Liping Ding, Data curation, Investigation, Methodology | Rui Liao, Data curation, Methodology, Software | Nini Li, Data curation, Formal analysis, Validation, Writing – original draft | Xiaoping Hong, Supervision, Writing – original draft, Writing – review and editing | Zhenyou Jiang, Validation, Visualization, Writing – original draft | Dongzhou Liu, Conceptualization, Funding acquisition, Supervision, Writing – review and editing

## ADDITIONAL FILES

The following material is available online.

### Supplemental Material

**Table S1 (Spectrum02826-23-s0001.xlsx).** All virus sequence IDs included in this study.

### Open Peer Review

**PEER REVIEW HISTORY (review-history.pdf).** An accounting of the reviewer comments and feedback.

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
