## [Reviewer comments · Microbiology Spectrum]

Microbiology Spectrum

Global genomic diversity and conservation of SARS-CoV-2 since the COVID-19 outbreak

Heng Li, Liping Ding, Rui Liao, Nini Li, Xiaoping Hong, zhenyou jiang, and Dongzhou Liu

Corresponding Author(s): Dongzhou Liu, Shenzhen People's Hospital

Review Timeline:

Submission Date:	July 14, 2023
Editorial Decision:	September 25, 2023
Revision Received:	September 26, 2023
Accepted:	September 27, 2023

Editor: Cyprian Rossetto

Reviewer(s): Disclosure of reviewer identity is with reference to reviewer comments included in decision letter(s). The following individuals involved in review of your submission have agreed to reveal their identity: LATA KUMARI (Reviewer #1); ASAAD MOHAMMED ATAA (Reviewer #2)

Transaction Report:

DOI: <https://doi.org/10.1128/spectrum.02826-23>

September 25, 2023

Prof. Dongzhou Liu
Shenzhen People's Hospital
1017 Dongmen North Road
Shenzhen
China

Re: Spectrum02826-23 (Global genomic diversity and conservation of SARS-CoV-2 since the COVID-19 outbreak)

Dear Prof. Dongzhou Liu:

Thank you for submitting your manuscript to Microbiology Spectrum. There is substantial information within the manuscript that will be of interest to those studying the dynamics of SARS-CoV-2 infection. Please see the reviewers notes below and address them in a modified manuscript, specifically the presentation of Table 1.

Link Not Available

Sincerely,

Cyprian Rossetto

Journals Department
Reviewer comments:

Reviewer #1 (Comments for the Author):

Table 1 can be formatted differently for easy understanding.

Reviewer #4 (Comments for the Author):

The manuscript describes the analysis of 80,833 sequences of SARS-CoV-2 genomes to characterize their diversity and conversion balances. The manuscript is generally technically sound and seems to address some of the research interests in

SARS-CoV-2.

Authors should address the comments below:

1. Line 44: The text should have a heading of background but this is missing.
2. Line 51-52: There are spaces in the text that need to be formatted.
3. Line 130: Authors should state the actual number of mismatches instead of just indicating that it is more than 40
4. Line 148-149: The symbol indicated in the text "和" is not explained. The authors should amend this.
5. Line 307: The "n" should be changed to "N". I guess this refers to the nucleocapsid gene.

Staff Comments:

Preparing Revision Guidelines

Please return the manuscript within 60 days; if you cannot complete the modification within this time period, please contact me. If you do not wish to modify the manuscript and prefer to submit it to another journal, please notify me of your decision immediately so that the manuscript may be formally withdrawn from consideration by Microbiology Spectrum.

1 **Global genomic diversity and conservation of SARS-CoV-2 since the COVID-19**
2 **outbreak**

3 Heng Li^{a,b,c,*}, Liping Ding^a, Rui Liao^a, Nini Li^d, Xiaoping Hong^a, Zhenyou Jiang^e,
4 Dongzhou Liu^{a*}

[revised manuscript text omitted]

ORF7a	BAG6	Virus-host interactome and proteomic survey of PMBCs from COVID-19 patients reveal potential virulence factors influencing SARS-CoV-2
	QPCTL	
	XPOT	
	PSMD1	
	AT2A2	
	PP6R3	
	COPA	
	PSMD2	
	PRS7	
	TECR	
	COPB	
	PSMD7	

AT1A1
 PRS4
 DNJA2
 UBR5
 PSD13
 PSMD6
 HSP74
 SC16A
 CMC1
 MPCP
 GEMI4
 PSA
 DNJC7
 RPB2
 PRS8
 GHC1
 CALU
 EMD
 DNJA3
 TXTP
 BAG2
 DDX20
 DJC10
 ECHA
 ATPB
 2AAA
 HACD3
 RO60
 XPO1
 COPG2
 COPB2
 M2OM
 FAF2
 QPCTL
 COPA
 AT2A2
 MIC60
 TECR
 SMC2
 XPOT
 TMM33
 COPB

ORF7b

Virus-host interactome and proteomic survey of PMBCs from COVID-19 patients reveal potential virulence factors influencing SARS-CoV-2

	MPCP	
	PSA	
	AT1A1	
	ATD3A	
	SND1	
	NOC3L	
	ATPB	
	DNJA1	
	ATPA	
	CALU	
	DPM1	
	RCN2	
	PLOD2	Virus-host interactome and proteomic survey of PMBCs from COVID-19 patients reveal potential virulence factors influencing SARS-CoV-2
	TOR1A	
	STC2	
	PLAT	
	ITGB1	
	CISD3	
	COL6A1	
	PVR	
	DNMT1	
	LOX	
	PCSK6	
	INHBE	
ORF8	NPC2	
	MFGE8	A SARS-CoV-2-Human Protein-Protein Interaction Map Reveals Drug Targets and Potential
	OS9	
	NPTX1	
	POGLUT2	
	POGLUT3	
	ERO1B	
	PLD3	
	FOXRED2	
	CHPF	
	PUSL1	
	EMC1	
	GGH	
	ERLEC1	
	IL17RA	
	NGLY1	

HS6ST2
SDF2
NEU1
GDF15
TM2D3
ERP44
EDEM3
SIL1
POFUT1
SMOC1
PLEKHF2
FBXL12
UGGT2
CHPF2
ADAMTS1
HYOU1
FKBP7
ADAM9
FKBP10
HACD3
BAG6
COPG2
XP01
SMC2
MIC60
COPA
AT2A2
CMC1
PSMD1
COPB
TECR
M2OM
XPOT
PRS7
UBR5
AT1A1
PSMD2
DNJA2
ATD3A
SND1
RPB2
SC16A

	MPCP	
	PSD13	
	PSMD6	
	DNJA1	
	GHC1	
	TF3C1	
	CALU	
	GEMI4	
	PSA	
	PSA7	
	GNAS1	
	ECHA	
	ATPA	
	PRS8	
	ATPB	
	DDX20	
	EMD	
	EIF3A	
	PPT1	Virus-host interactome and proteomic survey of PMBCs from COVID-19 patients reveal potential virulence factors influencing SARS-CoV-2

[revised manuscript text omitted]

518 2022, 185(5):860-871.e813.

Reviewer #1

Table 1 can be formatted differently for easy understanding.

Thank you very much for your comment. Table 1 has been redrawn to enhance readability and compress the layout. As shown:

SARS-CoV-2 proteins	Human interacting protein	References
ORF6	NUP98, RAE1, MTCH1, KINH, EDC4, DCTN2, ATD3A, RAE1L, FAF2, XP01, SPF27, MYCB2, AT1A1, ATD3B, AT2A2, ACLY, DESP, ATPB, RT27, ATPA, CE170	A SARS-CoV-2-Human Protein-Protein Interaction Map Reveals Drug Targets and Potential
	HEATR3	A SARS-CoV-2-Human Protein-Protein Interaction Map Reveals Drug Targets and Potential
ORF7a	MDN1, HACD3, FAF2, XPO1, COPG2, COB2, SMC2, MIC60, BAG6, QPCTL, XPOT, PSMD1, AT2A2, PP6R3, COPA, PSMD2, PRS7, TECR, COB2, PSMD7, AT1A1, PRS4, DNJA2, UBR5, PSD13, PSMD6, HSP74, SC16A, CMC1, MPCP, GEMI4, PSA, DNJC7, RPB2, PRS8, GHC1, CALU, EMD, DNJA3, TXTP, BAG2, DDX20, DIC10, ECHA, ATPB, 2AAA	Virus-host interactome and proteomic survey of PMBCs from COVID-19 patients reveal potential virulence factors influencing SARS-CoV-2
ORF7b	HACD3, RO60, XPO1, COPG2, COB2, M2OM, FAF2, QPCTL, COPA, AT2A2, MIC60, TECR, SMC2, XPOT, TMM33, COB2, MPCP, PSA, AT1A1, ATD3A, SND1, NOC3L, ATPB, DNJA1, ATPA, CALU, DPM1, RCN2	Virus-host interactome and proteomic survey of PMBCs from COVID-19 patients reveal potential virulence factors influencing SARS-CoV-2
	PLOD2	Virus-host interactome and proteomic survey of PMBCs from COVID-19 patients reveal potential

Reviewer #4

The manuscript describes the analysis of 80,833 sequences of SARS-CoV-2 genomes to characterize their diversity and conservation balances. The manuscript is generally technically sound and seems to address some of the research interests in SARS-CoV-2.

Authors should address the comments below:

1. Line 44: The text should have a heading of background but this is missing.

“BACKGROUND” has been added.

2. Line 51-52: There are spaces in the text that need to be formatted.

This is due to the layout display caused by a long URL, which actually only has one space in the local software.

Online:

51 and variants under monitoring (VUM)
52 (<https://gisaid.org/database-features/influenza-genomic-epidemiology/>). In recent

Local display:

51 under monitoring (VUM) (<https://gisaid.org/database-features/influenza-genomic-epidemiology/>). In recent years, there have been several rounds of mainstream strain

3. Line 130: Authors should state the actual number of mismatches instead of just indicating that it is more than 40

"there were 295 genomes with more than 40 mismatches" in Figure 4A indicates that on the left side of the graph, there are 295 genomes with mutation counts exceeding 40. The number of mismatches varies for each of the 295 genomes, and they are represented by orange bars in the graph. The top three genomes with the highest mutation counts have already been marked in the graph.

4. Line 148-149: The symbol indicated in the text "和" is not explained. The authors should amend this.

Modified.

two variants with the highest mismatches (MZ233993.1 and MZ097525.1) were highly similar in sequence, only two bases are different. C: MZ233993.1 and MZ097525.1

5. Line 307: The "n" should be changed to "N". I guess this refers to the nucleocapsid gene.

Modified.

of the start codon of the N protein, which may affect translational initiation.

September 27, 2023

Prof. Dongzhou Liu
Shenzhen People's Hospital
1017 Dongmen North Road
Shenzhen
China

Re: Spectrum02826-23R1 (Global genomic diversity and conservation of SARS-CoV-2 since the COVID-19 outbreak)

Dear Prof. Dongzhou Liu:

Thank you for the modification to your manuscript based on the reviewers comments. Your manuscript has been accepted, and I am forwarding it to the ASM Journals Department for publication. You will be notified when your proofs are ready to be viewed.

Sincerely,

Cyprian Rossetto
Editor, Microbiology Spectrum